# Promoting Individual and Organizational OCBs: The Mediating Role of Work Engagement

**DOI:** 10.3390/bs10090138

**Published:** 2020-09-14

**Authors:** Flavio Urbini, Antonio Chirumbolo, Antonino Callea

**Affiliations:** 1Department of Human Sciences, LUMSA University, 00193 Rome, Italy; a.callea@lumsa.it; 2Department of Psychology, Sapienza University of Rome, 00185 Rome, Italy; antonio.chirumbolo@uniroma1.it

**Keywords:** OCBs, work engagement, job satisfaction, mediation mechanism, social exchange theory

## Abstract

In today’s dynamic organizational environment, employees with a tendency to display discretional behaviors beyond their prescribed formal job duties represent a plus. Underpinned by the theories of social exchange and conservation of resources, these behaviors can be influenced by their level of job satisfaction (JS), defined as the extent to which employees like their work, and work engagement (WE), defined as a positive work-related state of mind. The present study investigates the mediating mechanism of WE in the relationship between JS and organizational citizenship behaviors (OCBs), which refer to discretionary behaviors that could benefit an organization (OCBs-O) and/or its members (OCBs-I). The mediational hypothesis is examined using structural equation modeling (SEM) among 719 Italian private and public sector employees. The significance of total, direct, and indirect effects was tested via bootstrapping. The results showed that JS was positively related to WE, which, in turn, was positively related to both OCBs-I and OCBs-O. The SEM results supported the hypotheses: WE fully mediated the relationship between JS and OCBs-I, and it partially mediated the relationship between JS and OCBs-O. This study sheds new light on this mechanism. Consequently, it is useful for HRM policy. It also helps us to better understand how satisfied and engaged employees are willing to adopt positive organizational behaviors.

## 1. Introduction

Due to external events and protracted economic crises, the constant and rapid changes in the world of work have forced both scholars and managers to try to better understand the variables that impact strategic organizational goals and enhance employee well-being in both the public and the private sector.

From an organizational point of view, a strategic asset clearly has a high level of performance related to the work. In addition to performance that is strictly related to the task—which is called task performance—there is another form of performance that is not formally required as part of the job, but which helps to shape the social and psychological context of the organization, which is called contextual performance [1]. The latter falls in the domain of employee performance, which is composed of activities; that is, a class of behaviors—called organizational citizenship behaviors (OCBs)—that go beyond the prescribed formal job duty. Due to their relevance, OCBs have gained momentum ever since researchers first coined the term in the early part of the 1980s [2,3]. OCBs include behaviors that are extra to the role, entirely voluntary, and not formally assigned or rewarded. At the core of OCBs are discretionary behaviors, which benefit organizations and their members by improving the social and psychological environment in which the technical core of the organization must function [1,4]. A multidimensional approach to studying OCBs has been proposed for review (see [5]). For example, Williams and Anderson [6] distinguished OCBs with respect to the target of the behavior’s beneficiary. They proposed a dichotomous model, which refers to organizational citizenship behaviors toward individuals (OCBs-I)—that is, behaviors that benefit other organizational members (including, for example, volunteering to help a co-worker)—and to organizational citizenship toward an organization (OCBs-O)—that is, behaviors that benefit the organization through, for example, praising the organization to outsiders.

Considering that all organizations are, to some degree, dependent on extra-role activities, those behaviors that may improve individual and organizational efficiency and that also increase performance become more valuable. In this respect, understanding the variables that impact the employee’s OCBs as critical drivers of organizational effectiveness is an obvious concern for organizations. The importance of OCBs to productivity and increased efficiency [7] has led to numerous attempts to identify their antecedents at individual and group levels [8].

The present study represents an effort to gain a deeper understanding of the effect of job satisfaction (JS) and work engagement (WE) on the dimensions of OCBs. More specifically, the current paper aims to contribute to this growing area of research in five ways. First, this study enhances the research on individual-level antecedents of OCBs by investigating their relationship with JS, which will hopefully shed more light on the controversial findings in the existent literature for a review (see [8]). In fact, this study provides new empirical evidence advancing a model that integrates findings in prior studies, suggesting a novel model that simultaneously includes JS, WE, OCBs-I, and OCBs-O. Second, this study contributes to the debate about the beneficial effects of OCB dimensions by exploring WE as a potential mediator in the relationship of the JS-OCB dimensions. Third, the current study is the first empirical study to investigate the relationship between JS and OCB dimensions via WE as mediator in the same model. It also considers both dimensions of OCBs, toward the individual and the organization. Fourth, this study investigates the dimensionality of the OCB construct, and it validates the utility in the distinction between OCBs-I and OCBs-O for a meta-analysis (see [9]). Finally, this study is based on a well-established taxonomy for examining OCBs, that is, OCBs-I and OCBs-O [6]. This suggests that further investigation should evaluate the different facets of OCBs.

Grounded on the social exchange theory (SET) and on the conservation of resources (COR) theory, the current paper argues that JS can result in more positive OCBs toward individuals than towards organizations through WE.

## 2. Theoretical Framework and Hypotheses

### 2.1. Job Satisfaction as an Antecedent of Work Engagement and OCBs in the SET

The theoretical support for this study was built on the basis of the SET [10] and COR theory [11] to explain how JS influences OCB dimensions through WE. The SET perspective suggests that employees within an organization are motivated to reciprocate perceived favors that are provided by the organization. According to Blau [10], the norm of reciprocity through voluntary actions of individuals that are motivated by the returns that they are expected to bring from others is the core of moving the relationship forward. In other words, a relationship that is built upon social exchange embeds a mutual implicit agreement following a “do ut des” ratio.

Consequently, we argue that social exchange influences the willingness to practice OCBs for those employees with high levels of JS. When workers are more satisfied in their jobs and, hence, feel the need to reciprocate this positive work condition, they tend more to practice OCBs-I and OCBs-O.

JS can generally be defined as how fond a worker is of their job [12], the extent to which employees like their work, a positive emotional state that is based on an evaluation of job experience [13], and so on. In essence, JS refers to an employee’s affective evaluation, from pleasurable to unpleasurable, of the own job situation based on the extent to which the work environment fulfills their expectations, needs, values, or personal characteristics [14,15]. Even though there have been many definitions, to date, no single definition or theory can cover the full spectrum of JS aspects [16]. Furthermore, JS can also be discussed in global or facet aspects [12]. For example, the facet perspective refers to the extent to which workers feel positively or negatively toward the intrinsic and/or extrinsic aspects of their job [17,18]. Intrinsic job facets are focused on the chance to use, for example, personal abilities, job participation, and involvement, while the extrinsic facets place emphasis on the material elements of the job, such as pay and career advancement [19]. This study will only use the intrinsic facet because it could be a motivational force [20], thus improving work attitudes, such as WE and, consequently, some activities, such as OCBs.

JS directly affects OCBs, and it can explain some portions of these behaviors [3,21]. Starting from the 1970s, considerable attention has been given to the causal relationship between JS and OCBs. However, these studies have failed to establish a conclusive link [8]. For example, many scholars consider that JS directly affects OCBs [22,23], whereas others did not find a direct association between them [24,25]. Furthermore, some studies have found differences in the relationship between JS and the OCB dimensions [26,27,28]; on the other hand, others did not find them [9]. Our position is much like that of William and Anderson [6] in that we considered OCBs as a bi-dimensional construct that is composed of OCBs-I and OCBs-O—with different antecedents—and it results in relationships between variables such as JS. Following this line of reasoning, JS can be the cause of both dimensions of OCBs [29]. According to Zeinabadi [30], this happens because satisfaction with the different aspects of a job encourages positive behaviors among employees. Methodologically speaking, relationships may differ across dimensions, and some dimensions may have independent effects on the same antecedent. For example, a distinction can be made between a satisfied employee who tends to be very helpful and cooperative toward others (OCBs-I), and yet, at the same time, tends to be certain about getting involved in the decisions that affect the organization (OCBs-O). Accordingly, we propose the following hypothesis:

**Hypothesis** **1.**
*JS will be positively related to OCBs-I (H1a) and OCBs-O (H1b).*


Most scholars agree that JS and WE are distinct variables [31,32]. Indeed, JS is a passive and affective state, whereas WE is an active state, and its content is strictly related to work [14]. More specifically, JS is a function of perceptions and affect toward the job, and the worker may or may not feel satisfaction compared to a specific component of the job. In contrast, WE is defined as a work-related state of mind that is characterized by vigor, dedication, and absorption [33]. Hence, it can be placed in the field of occupational health psychology [34] because it is assumed to be a strictly positive and relatively stable indicator of occupational well-being [33].

From a theoretical point of view, COR may explain the relationship between JS and WE. The primary tenets of COR theory are that individuals strive to protect and sustain—that is, to conserve—their current pool of resources that are valuable to them (e.g., objects, personal characteristics, job conditions, and energies) [35,36]. The resources lost have a greater psychological impact than the resources gained because they are related to the stress process. In the working context, resources may include time for work, status at work, adequate income, job training, and personal energy [35]. In order to conserve JS, workers may be more engaged. As a matter of fact, some studies suggested that engaged employees are potentially more satisfied with their jobs [14,37]. In the current study, JS is considered as the antecedent of WE. Based on prior research, which has found support for the centrality of satisfaction to engagement [31,32,38,39,40], we propose the following hypothesis:

**Hypothesis** **2.**
*Job satisfaction will be positively related to work engagement.*


### 2.2. The Mediating Role of Work Engagement

In the literature, WE has been found to be positively associated with contextual performance. There are several reasons for this, including experiencing positive emotions, enjoying a high level of psychological and physical health, and spreading engagement among co-workers [41]. Engaged employees are likely to perform activities that are not part of their formal job descriptions, but which nevertheless promote organizational effectiveness [42,43]. WE is thought to be an indicator of an employee’s willingness to expend discretionary effort to help the employer [44], showing a positive attitude and behavior within the work environment [45]. From a COR theory perspective, WE can be conceptualized as having an abundance of energy for, attachment to, and motivation for one’s work [46]. According to the COR theory, the JS–OCB-dimension relationship may be enriched considering that JS is a valuable resource for positive job energy. In turn, this positive job energy enhances the likelihood that employees are more engaged, triggering a positive relationship between JS and OCBs-I and OCBs-O. That is, the translation of the positive energy generated from satisfaction with the job into OCBs should be enhanced to the extent that employees have a positive work-related state of mind, i.e., are engaged in their work.

More explicitly, according to the COR theory, we argue that WE acts as a mediator that links JS to both OCB dimensions. Specifically, when employees are satisfied with their jobs, they feel a sense of engagement in terms of self-investment, energy, and passion, which, in turn, may produce positive extra-role behaviors, such as practicing OCBs-I and OCBs-O.

The positive relationship between JS and both OCBs-I and OCBs-O may be mediated by the employee’s WE because JS may not always result in productivity [14], while engagement seems to be a reliable predictor of contextual performance [46,47]. In addition, engaged employees are more activated toward performing better and behaving positively in the workplace [48], and are hence willing to practice OCBs.

Consistently with the COR theory, employees’ JS should spur their WE levels, and this effect could explain that employees’ residual energy at their disposal is allocated to OCBs toward their organizations and/or toward their coworkers in order to maintain their own positive resources.

To our knowledge, just one study has considered WE as a mediator in the linkage between JS and OCBs in a health workers sample, showing a partial mediation between these variables [49]. Following these results, in this model, we included both OCB dimensions, and we consider this to be multidimensional variable because the behavioral dimensions seem to be conceptually distinct; therefore, it may be appropriate to consider them separately. We postulate that WE refers to the relationship between JS and both OCBs-I and OCBs-O, and constitutes such a bridge. More specifically, the present study suggests that the relationship of JS and OCB dimensions is differently mediated by WE.

There are currently no specific studies analyzing the mediational role of WE in the relationship between JS and OCB dimensions. Following this lack of literature, we specifically investigated WE as a mediator.

Therefore, we propose the following hypotheses:

**Hypothesis** **3.**
*WE will be positively related to OCBs-I (H3a) and OCBs-O (H3b).*


**Hypothesis** **4.**
*WE will mediate the positive relationship between JS and OCBs-I (H4a) and OCBs-O (H4b).*


Figure 1 illustrates the conceptual model of the proposed mediation model that we used to test our hypotheses.

## 3. Research Methodology

### 3.1. Sample and Procedures

A total of 719 Italian employees participated in the present study in the years preceding the COVID-19 pandemic. Participants (21.2% males and 78.8% females) ranging from 19 to 67 years old (*M* = 38.42, *SD* = 10.91), worked in public (52.7%) or private (47.3%) organizations, and had permanent (50.7%) or fixed-term (49.3%) contracts. On average, the participants had worked in their organizations for about 10 years. Most of them (92.6%) were single (or lived alone), and the rest lived together (7.4%). Regarding education, 62.1% had high-level education (i.e., university degree or more), while 37.9% had a low-level education (i.e., compulsory degree). The participants completed a self-report questionnaire in paper format (60.5%) or via computer-assisted web interviewing (39.5%). All of the participants were unpaid volunteers. Their written informed consent was requested and their privacy was guaranteed.

The sample was selected via a snowball procedure, with a main concern (due to this non-probabilistic procedure) of obtaining a heterogeneous and varied sample in terms of socio-demographical features.

### 3.2. Measures

The self-report questionnaire contained socio-demographic information and the following measures.

#### 3.2.1. Job Satisfaction

JS was measured by the Italian short version of the Minnesota Satisfaction Questionnaire (MSQ) [50], adapted from the original version from Weiss et al. [18]. This scale measure two dimensions of JS: intrinsic (i.e., peculiar aspects related to the nature of the work, with the job tasks themselves, and as internally experienced by the worker) and extrinsic (i.e., aspects not related to the nature of the work and mainly under the employer’s control). For the purposes of this study, only the former dimension was used. The Italian short version consisted of 12 items, and it asks participants what they think about various aspects of their jobs, ranging from 1 (very dissatisfied) to 5 (very satisfied) (sample item: “The feeling of accomplishment I get from the job”). In the present study, the scale’s reliability was high (Cronbach’s alpha = 88).

#### 3.2.2. Work Engagement

WE was measured by the Italian validation [51] of the short Utrecht Work Engagement Scale (UWES-9) [52]. This scale investigated the positive aspects of vigor, dedication, and absorption at work. The scale consisted of nine items, and it asked the participants about how often they have feelings related to work, ranging from 0 (never) to 6 (always) (sample item: “At my work, I feel that I am bursting with energy”). Following the recommendation of Schaufeli et al. [52], an overall engagement score of the UWES-9 was calculated in the analyses. In the present study, the scale’s reliability was high (Cronbach’s alpha = 87).

#### 3.2.3. Organizational Citizenship Behaviors

OCBs were measured with Lee and Allen’s [53] scale. This scale measure two dimensions of OCBs: OCBs toward individuals/co-workers and OCBs toward the organization. The scale consisted of 16 items, and it asked the participants to express their agreement with statements about their behaviors at work, ranging from 1 (strongly disagree) to 7 (strongly agree) (sample item OCBs-I: “Assist others with their duties”) (sample item OCBs-O: “Keep up with developments in the organization”). In the present study, the scale’s reliability was high for both OCBs-I (Cronbach’s alpha = 87) and OCBs-O (Cronbach’s alpha = 88).

#### 3.2.4. Control Variables

Some demographic and job-related characteristics that might covary with JS were controlled, such as: gender (0 = male; 1 = female), age (years), tenure (years), and contract type (0 = permanent contract; 1 = temporary contract).

## 4. Results

### 4.1. Data Analyses

The data were analyzed using SPSS (21th version) and the Mplus program (8.53 version). Preliminary analyses (i.e., Pearson correlations) were performed to explore the relationships between the control variables, JS, WE and organizational citizenship behaviors toward individuals (OCBs-I) and organizations (OCBs-O).

To test common method bias, four models were compared: (M1) the hypothesized four-factor model (i.e., JS, WE, OCBs-I, and OCBs-O); (M2) a three-factor model where the dimensions of OCBs were replaced by a single factor (i.e., JS, WE, and OCBs); (M3) a two-factor model in which JS and WE were replaced in a single factor, as were the dimensions of OCBs (i.e., JS, WE, OCBs); and (M4) a model in which all items were loaded on a single factor. To examine the divergent validity of the constructs, the four-factor measurement model (M1) was compared to other models via the χ^2^ difference test (Δχ^2^).

Mediation analyses with latent variables were performed via structural equation modeling (SEM) using two random composites of items (parcels) as indicators of each latent variable (e.g., Little et al. [54,55]). The item parcels were randomly selected, but contained a balanced number of items with comparable reliabilities. The mediation analysis strategy recommended by James et al. [56] was followed. In the first step, the full mediation model (i.e., without the direct effects) representing the best choice of a baseline model was tested. In the second step, the partial mediation model (including the direct effects from JS to OCBs-I and OCBs-O) was tested. The two nested models were compared via the χ^2^ difference test (Δχ^2^). When Δχ^2^ was not significant, the full mediation model had to be preferred because it is more parsimonious. The better model was also compared with an alternative model that included the effects of control variables (e.g., age and gender). The two alternative models were evaluated by comparing Akaike’s information criterion (AIC). A lower AIC suggests a better and more parsimonious model. The SEM approach provides model fit information about the consistency of the hypothesized mediational model with the data, and it also provides evidence for the plausibility of the causality assumption. The goodness of fit of the model was evaluated with the χ^2^ statistic along the following indices: the comparative fit index (CFI), the Tucker–Lewis Index (TLI), the root mean squared error of approximation (RMSEA), and the standardized root mean square residual (SRMR). With respect to the interpretation of fit indices, several authors have considered that a model is reasonably good if the CFI and TLI are higher than 0.90, if the RMSEA and SRMR are lower than 0.08, and if χ^2^ is not significant [57,58]. Regarding the CFI and TLI indices, a more stringent threshold of 0.95 has been suggested to be preferred [59].

Finally, bootstrapping [55] with 5000 samples with replacement from the full sample to construct bias-corrected 95% confidence intervals (CI) was used to evaluate the significance of total, direct, and direct effects. The effects are significant when zero is not included between the lower-limit confidence intervals (LLCI) and the upper-limit confidence intervals (ULCI). Bootstrapping is one of the more valid and powerful methods for testing mediating variable effects [60] because it does not impose the assumption of normality of the sampling distribution. Regarding the word “effect” used earlier, it may suggest that there is a causal relationship, but (given the cross-sectional nature of the data) it does not make inferences about causality. This terminology is used simply for reason of clarification [61].

#### 4.1.1. Descriptive Statistics

Table 1 shows internal consistencies, means, standard deviations, and correlations between variables. First, Table 1 shows that the dimensions reached very good internal consistency reliability, with Cronbach’s α coefficients of between 0.87 (WE and OCBs-I) and 0.88 (JS and OCBs-O). Second, as expected, JS was positively related to WE and both dimensions of OCBs. Moreover, WE was positively related to OCBs-I and OCBs-O.

With respect to the control variables, gender was positively related to age and WE; age was negatively related to contract type and was positively related to tenure, JS, WE, OCBs-I, and OCBs-O; tenure was negatively related to contract type, OCBs-I, and OCBs-O; and contract type was negatively related to JS, OCBs-I, and OCBs-O.

#### 4.1.2. Measurement Model

With respect to common method bias, a comparison of the alternative models (see Table 2) suggested that the hypothesized model (M1) was a better fit to the data than other competitive models. Due to these results, the more parsimonious model (i.e., M1) had to be preferred. Furthermore, M1 reached good fit indices: χ^2^ (21) = 81.692, *p* < 0.00; CFI = 985; TLI = 975; RMSEA = 0.063; SRMR = 0.022. Moreover, the hypothesized four-factor model was used to test the mediational hypotheses.

#### 4.1.3. Mediational Model

The mediational hypotheses were tested by SEM, using maximum likelihood as the estimation method. The full mediation model was compared to an alternative partial mediation model via Δχ^2^. The full mediation model showed a satisfactory fit (χ^2^ (23) = 94.930, *p* < 0.00; CFI = 0.98; TLI = 0.97; RMSEA = 0.06; SRMR = 0.03), with AIC = 17183.763. The partial mediation model, including direct effects from JS on OCBs-I and OCBs-O, showed the following fit indices: (χ^2^ (21) = 81.692, *p* < 0.00; CFI = 0.98; TLI = 0.97; RMSEA = 0.06; SRMR = 0.02), with AIC = 17174.525. Both models showed acceptable fit indices. The difference between the two models was statistically significant Δχ^2^ (2) = 13.238, *p* < 0.01. Therefore, the partial mediation model had to be preferred.

This model was compared with a partial mediation model including control variables. The effects of the control variables on WE, OCBs-I, and OCBs-O were not significant, and the AIC = 18195.865. Because the AIC was higher, the partial mediation model without control variables was preferred to test the mediational hypotheses (see the model in Figure 2).

The SEM results show that the total effects of JS on OCBs-I (0.368, with bootstrap confidence intervals (CIs) between the lower limit (LL) of 0.293 and upper limit (UL) of 0.439) and on OCBs-O (0.466, with bootstrap CIs between the LL of 0.398 and UL of 0.529) were significant and positive, which supports H1a and H1b. Furthermore, JS was positively related to WE (0.706), which supports H2. Additionally, WE was positively and significantly related to OCBs-I (0.283) and OCBs-O (0.371), which supports H3a and H3b.

Finally, the results of the partial mediation model showed that the direct effect of JS on OCBs-I was not statistically significant (0.169, with bootstrap CIs between the LL of 0.058 and UL of 0.283), while the indirect effect via WE was significant and positive (0.199, with bootstrap CIs between the LL of 0.118 and UL of 0.279). This shows that WE fully mediated the relationship between JS and OCBs-I. Consequently, these results support H4a. Meanwhile, the direct effect of JS on OCBs-O was statistically significant (0.204, with bootstrap CIs between the LL of 0.105 and UL of 0.304), as was the indirect effect via WE (0.262, with bootstrap CIs between the LL of 0.189 and UL of 0.329). Therefore, WE partially mediated the relationship between JS and OCBs-O, supporting H5.

## 5. Discussions

This study explored the relationships among JS, WE, and extra-role performance, that is, OCBs toward individuals and toward the organization. To provide innovative results, this study tested if WE mediates the relationship between JS and two dimensions of OCBs (i.e., OCBs-I and OCBs-O). Many studies support the positive relationship between JS and OCBs. However, although the theoretical rationale for a mediation by WE is solid, this mediation has been tested only once [49], considering OCBs as a global construct. Overall, our results support the hypotheses. JS was positively related to OCBs-I and OCBs-O, which supports hypotheses 1a and 1b, respectively. The same is true for the relationship between JS and WE, which supports hypothesis 2. With respect to hypotheses 3a and 3b, the new and most important result of our study, compared to the results of Ng et al. [49], is that the two indirect (mediated) effects of WE on OCBs-I and OCBs-O were statistically significant. Consonant with hypotheses 4a and 4b, our results were as expected. WE mediated both the relationship between JS and OCBs-I, and also the relationship between JS and OCBs-O.

These results indicate that WE is an important link between JS and behaviors at work directed especially toward individuals compared with those toward the organization. In terms of extra-role behaviors, this means that engaged employees are supposed to be more likely to create a positive social context and to express emotional- and relational-oriented behaviors toward colleagues [7].

Our finding of complete and partial mediation suggests that WE is a central variable that transmits positive effects of JS into OCBs-I. In addition, there is not much variance left in the association between JS and OCBs-I that is unexplained by WE. With respect to the mediation mechanism of WE on OCBs-O, we expected partial mediation because we assumed the existence of other variables related to the organization (e.g., organizational identification and perceived organizational support), which could mediate the effect of JS on OCBs-O independently of WE. However, there is good reason to consider that these conclusions are only a tentative interpretation of the findings. Clearly, further research is needed to shed light on the issue of partial versus full mediation with variables strictly related to organizational context. However, the main issue is not only the distinction on whether mediation is partial or full, but also whether mediation via WE exists at all. Indeed, the most significant contribution of the present study is to inform on the difference of WE as a mediator in the linkage between JS, OCBs-I, and OCBs-O grounded in the SET and COR theory. This is especially important because, so far, just one study has tested this mediation, and only with an overall OCB evaluation [49].

In terms of theoretical contributions, the present study is the first empirical effort to explain the relationship between JS, WE, OCBs-I, and OCBs-O in a mediation model. As previous research investigated the relationships between the variables of this study separately, providing fragmented results, this study proposed an integrated view of them. Moreover, in order to fill the gap in the empirical evidence concerning JS–OCBs dimensions, we suggested and then empirically examined them through a cross-sectional study using a mediation model, hypothesizing the mediator role of WE in this relationship. The hypothesized model was built on the SET and COR theory as theoretical frameworks. Regarding the use of these theoretical perspectives, strong theory leads to good models and better-justified mediational inferences [62].

As the relationship between JS and OCBs is not automatic [8], this study extended current knowledge about the underlying mechanism, extending what we know about this relationship. An important mechanism that emerged in this relationship, according to the logic of the COR theory, is that resource gains from engaged employees have been translated into citizenship behaviors toward individual coworkers. Therefore, the COR theory may explain the indirect effects of JS on OCB dimensions: In order to preserve JS, workers may be incentivized to be more engaged and, consequently, to practice positive behaviors toward colleagues and their organizations. The empirical evidence in our study supports and extends previous research examining the dimensionality of the OCB construct, especially the distinction between OCBs-I and OCBs-O [6].

In line with this viewpoint, it has been demonstrated that voluntary helping behaviors at work as responses to positive feelings about the job depend also on feelings of energy, enthusiasm, and absorption in one’s work.

There are also some practical implications of the present study. Modern organizations that wish to stay competitive need engaged employees, who are available to perform activities and who also go beyond the prescribed formal job duty. Employees who are satisfied and are full of energy and enthusiasm are presumably more inclined to have OCBs-I and, therefore, altruism, courtesy, peacekeeping, cheerleading, and, generally, behaviors that are aimed at helping other individuals [6]. Given the linkage between WE and OCBs-I, HR departments should attempt to support and cultivate WE in their workforce. Practically speaking, WE has a strategic importance in the working environment because it represents a stable and genuinely positive disposition toward work and a work-related mood [34]. Therefore, HR specialists might be able to increase engagement by designing optimal working environments for employees. In addition, HR can find out which demands and resources need attention in terms of positive behaviors at work, and they can then propose and promote procedures accordingly.

Our findings emphasized the challenge that HR departments should face in stimulating voluntary helping behaviors within organizations. Organizations should be proactive and take the initiative to detect which circumstances make employees satisfied and engaged by analyzing the sources of such attitudes (e.g., adequate workloads, met job expectations).

These results highlighted the importance of maintaining high levels of the variables investigated. More specifically, HR departments can maintain and even enhance employees’ engagement, as well as productivity and, subsequently, well-being, by carefully balancing the effort to monitor employees’ day-to-day dynamics with the effort to promote a generally supportive climate in the work unit (e.g., by providing supervisory support with open communication).

In addition, training employees on the personal benefits of OCBs without making OCBs a mandatory requirement of the job can be used as an inexpensive and feasible way to encourage employees to perform OCBs within the organizational context and to enhance the employees’ well-being. Furthermore, organizations should use practices for all employees within the organization—from newcomers to top managers. The latter should promote a WE culture as a part of a chain that contributes to OCBs, and, in turn, to well-being at work. According to Bakker and Demerouti [63], organizations can use two approaches—top-down and bottom-up—to facilitate WE. The former includes initiatives at the HR level, such as leadership interventions, whereas the bottom-up approach refers to various strategies of training individual proactiveness, thus increasing the personal resources of employees, such as optimism, self-efficacy, and self-esteem because they help optimize the work environment in terms of affordable job demands and sufficient job resources. In line with the SET, when employees feel that they are treated well by their organizations, also in terms of taking care of their well-being, they reciprocate and exceed the minimum of the job requirements and required duties, maximizing citizenship behaviors to help others and the organization. According to the COR theory, positive energy stemming from employees’ current job satisfaction fuels more their engagement and, in turn, the propensity to go out of their way to help their coworkers voluntarily. Thus, WE is an important concept for practitioners to understand the impact of OCBs in the expansion of internal organizational strategic processes and for the employees’ well-being.

Several limitations of the present study should be acknowledged. First, our data were collected at one point in time using a cross-sectional design. However, this design cannot support claims of causality and it cannot exclude the possibility of reversed mediation. However, it should be noted that our theorizing of the hypotheses followed the existing literature as closely as possible. Future longitudinal research using a multi-wave design should further clarify the directions of the relationships between the variables that we studied.

Second, because all of the variables regarding both predictor and criterion in this study were self-reported by the respondents, the possibility of inflated relationships among these measures due to the common method variance bias is possible. To control these method biases, the following strategies were adopted. We conducted a single-factor confirmatory factor analysis (CFA) test and identified the structure of the measure before testing the hypotheses. Although this remedy cannot completely solve that issue, it dispels the doubt that our conclusions could be overthrown by the bias. We used a counterbalancing order of the measurement for the predictor and control variables. Furthermore, in reference to self-reported measures of the OCB dimensions, as part of contextual performance, it would be desirable in future research to obtain ratings from other sources (e.g., colleagues, peers, or superiors) to solidify the present research’s findings and exclude the potential single-method bias. However, we deliberately chose self-rate evaluation of OCBs because these ratings are inherently subjective, while ratings of a person’s own OCBs toward individuals and organizations are a poor substitute for independent judgments.

Third, the procedure of data collection via the snowball procedure could be criticized for the lack of representativeness and selection bias problem. We used this procedure to increase the willingness to participate in this study. Regarding the sample, our results clearly underscore the importance of WE as a mediator between JS, OCBs-I, and OCBs-O, but further research should involve different independent samples suggesting that these mediations, partial and full, do exist. In addition, mediation could be found with different scales to measure OCBs, underscoring the stability of the effects. However, future studies may verify the generalizability of these results in other samples. In this respect, only workers from the Italian public and private sectors participated in the present study. It would be interesting to validate the present results with other populations.

Finally, even though our results indicate the importance of WE, we cannot exclude that alternative explanations may account for the mediation between JS and OCBs through WE. There might be a third variable that is related to JS, WE, and OCBs. For example, it may be possible that an employee with high levels of JS has a high level organizational identification and, as a consequence of the positive relationship with the organization, has more energy and enthusiasm (i.e., WE) and accomplishes more OCBs-I and OCBs-O. Further research should take such alternative explanations into account.

## 6. Conclusions

To recap, our study extends previous research by showing the mediating role of WE in the relationship of JS with OCBs-I and OCBs-O on the one hand, and the strategic importance of promoting individual and organizational OCBs on the other hand. Finally, we highlight our concern that comparing the roles played by JS and WE in predicting any set of behaviors at work is a challenging and complex endeavor. Indeed, it might never be possible to precisely sort out the relative contributions of these two constructs (or any) to the OCB dimensions. Nonetheless, we believe this to be an important and consequential issue, and we look forward to future research that attempts to shed additional light on it with the final aim of increasing both the employees’ well-being and the organizational productivity.

## Figures and Tables

**Figure 1 behavsci-10-00138-f001:**
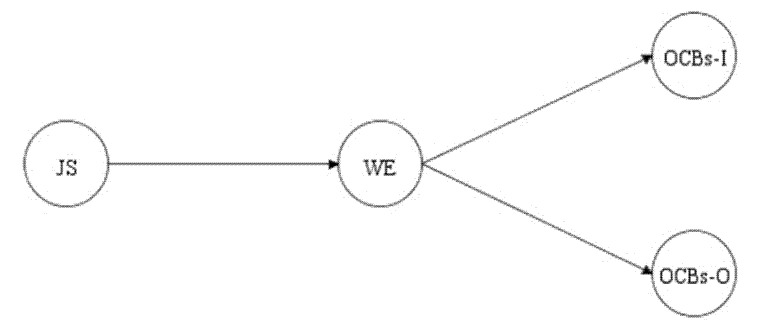
The hypothesized mediational model. Note: JS = job satisfaction; WE = work engagement; OCBs-I = organizational citizenship behaviors directed toward individuals; OCBs-O = organizational citizenship behaviors directed toward the organization.

**Figure 2 behavsci-10-00138-f002:**
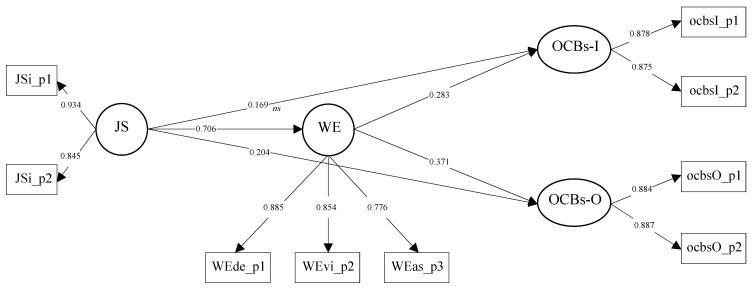
The partial mediational model with standardized coefficients. Note: Jsi_p1 = Parcel 1 of JS; Jsi_p2 = Parcel 2 of JS; JS = job satisfaction; Wede_P1 = Parcel 1 of WE (dedication); Wevi_P2 = Parcel 2 of WE (vigor); Weas_P3 = Parcel 3 of WE (absorption); WE = work engagement; ocbsI_p1 = Parcel 1 of OCBS-I; ocbsI_p2 = Parcel 2 of OCBS-I; OCBS-I = organizational citizenship behaviors directed toward individuals; ocbsO_p1 = Parcel 1 of OCBS-O; ocbsO_p2 = Parcel 2 of OCBS-O; OCBs-O = organizational citizenship behaviors directed toward the organization.

**Table 1 behavsci-10-00138-t001:** Means, standard deviations, internal consistencies, and correlations among variables.

	M	SD	2	3	4	5	6	7	8
1. Gender	-	-	0.10 *	0.04	−0.03	0.00	0.13 **	0.08 *	0.03
2. Age	38.42	10.91	-	0.76**	−0.61 **	0.19 **	0.15 **	0.20 **	0.16 **
3. Tenure	9.05	9.42		-	−0.55 **	0.16 **	0.04	0.10 **	0.79
4. Contract type	-	-			-	−0.17 **	−0.04	−0.10 *	−0.13 **
5. JS	3.49	0.68				(0.88)	0.61 **	0.34 **	0.42 **
6. WE	4.61	1.33					(0.87)	0.36 **	0.46 **
7. OCBs-I	4.72	1.30						(0.87)	0.62 **
8. OCBs-O	4.51	1.41							(0.88)

Note: JS = job satisfaction; WE = work engagement; OCBs-I = organizational citizenship behaviors directed toward individuals; OCBs-O = organizational citizenship behaviors directed toward the organization. Gender: 0 = male; 1 = female; contract type: 0 = permanent contract, 1 = temporary contract. Cronbach’s α values are in parentheses. * *p* < 0.05; ** *p* < 0.01.

**Table 2 behavsci-10-00138-t002:** Confirmatory factor analysis (CFA) and comparison of different measurement models.

Model	CFI	TLI	RMSEA	SRMR	*χ* ^2^	Df	Δ*χ*^2^ (M…−M_1_)	Δdf (df…−df_1_)
M1	0.985	0.975	0.063	0.022	81.692	21	-	-
M2	0.910	0.866	0.146	0.046	392.516	24	310.824 **	3
M3	0.821	0.753	0.198	0.066	761.199	26	607.507 **	5
M4	0.609	0.479	0.288	0.131	1635.430	27	1553.738 **	6

Note: M1 = four factors: JS, WE, OCBs-I, and OCBs-O; M2 = three factors: JS, WE, and OCBs-I+OCBs-O; M3 = two factors: JS+WE and OCBs-I+OCBs-O; M4 = no measurement discrimination; ** *p* < 0.001.

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
