# Peer review of "Promoting Individual and Organizational OCBs: The Mediating Role of Work Engagement"

_behavsci, 2020, doi:10.3390/bs10090138_

Round 1

Reviewer 1 Report

I consider the paper interesting especially in this historical phase where organizations have to redesign their working models.

I believe that the paper is written in accordance with the requirements for the scientific study, therefore the following comments should be treated as aimed at improving it and increasing the readability of the text and the quality of the presentation of the results.

The theoretical context is well structured and well argued, however, the studies cited date back to somewhat outdated research, it would be appropriate to update the literature on the topic. For example:

A Study on the Relationship between Employee Engagement and Organizational Citizenship with Reference to Employees Working in Travel Organizations
doi:10.12727/ajts.14.3

Job satisfaction and organizational citizenship behaviour amongst health professionals: The mediating role of work engagement
https://doi.org/10.1080/20479700.2019.1698850

Person–environment fit at work:Relationships with workplace behaviours
https://journals.sagepub.com/doi/10.1177/1038416219870205

In addition, authors should highlight the fact that previous research lacks empirical studies to explain the relationships between OCBs-I, OCBs-O, JS and WE. This would further enhance the originality of the work.

I also invite the authors to specify the meaning of the acronyms used already from the abstract, to make it easier to read for those who do not have this specific sector-specific jargon.

Although the statistical model is well described from a methodological point of view in relation also to its ability to explain the hypotheses, the conclusions are perhaps too concise. It would be appropriate to better explain the theoretical and managerial implications of the results obtained in the study.

I encourage authors to improve their manuscript to make it suitable for publication.

Reviewer 2 Report

This is a nicely written paper, but it needs development on the following lines:

  1. The authors claim the novelty of their study, but the contribution of the paper is not clear. The reason is: the authors have not properly taken stock of the previous literature to position themselves. The hypothesized relationships have been tested in bits and pieces in several of the previous studies, so the authors should create a clear niche for themselves by clarifying how their study makes a theoretical contribution.
  2. The authors have used SET as a theoretical lens to explain the relationship between JS and OCBs, which needs further clarification in terms of how reciprocity/social exchange works in this relationship.
  3. The authors do not provide a theoretical basis for how JS fosters WE, thus the connection suggested is vague.
  4. Since SET does not seem to offer a strong basis for the relationships in the model, the authors should consider other theories such as conservation of resources theory and job resources model to enrich the rationale for the relationships proposed.
  5. One contribution could be to use two different theories for direct relationship of JS with OCBs and another via WE. This way the authors could show the incremental validity of one theory above the other. For example, SET might be use for direct relationship, and conservation of resources theory for indirect relationship. Previous literature suggests that JS helps in conserving resources and employees with more personal resources are better engaged and they perform well.

Round 2

Reviewer 1 Report

Dear authors,

the work you have done to improve your paper is very welcome.
Now the reading is easier thanks to the integrations you have added.
Therefore, in this new version, I consider the manuscript suitable for its publication in the journal.

Kind regards.